# The Characteristics of Multilocus Sequence Typing, Virulence Genes and Drug Resistance of *Klebsiella pneumoniae* Isolated from Cattle in Northern Jiangsu, China

**DOI:** 10.3390/ani12192627

**Published:** 2022-09-30

**Authors:** Tianle Xu, Xinyue Wu, Hainan Cao, Tianxu Pei, Yu Zhou, Yi Yang, Zhangping Yang

**Affiliations:** 1Joint International Research Laboratory of Agriculture and Agri-Product Safety, Ministry of Education of China, Yangzhou University, Yangzhou 225009, China; 2College of Animal Science and Technology, Yangzhou University, Yangzhou 225009, China; 3College of Veterinary Medicine, Yangzhou University, Yangzhou 225009, China

**Keywords:** *Klebsiella pneumoniae*, multilocus sequence typing, virulence gene, drug resistance

## Abstract

**Simple Summary:**

In this study, we used *K. pneumoniae* isolated from the milk of dairy cows with clinical mastitis from Jiangsu Province to investigate the distribution of virulence gene expression and test their antimicrobial susceptibility with defined MLST. The potential links in between virulence genes expression and antimicrobial resistance was also uncovered in a phenotypic way. The study provides insight into the spread of infection by *K. pneumoniae* and resistance in the case of bovine mastitis, as well as the potential target genes involved in the links between virulence determinants and MDR.

**Abstract:**

*Klebsiella pneumoniae* (*K. pneumoniae*) induced bovine mastitis has been becoming one of the dominantly pathogenic bacteria in cases of bovine mastitis, and is threatening public health through dairy products. In order to explore the characteristics of multilocus sequence typing (MLST), virulence gene carrying, and the relationship between virulence genes and the antibiotic resistance of *Klebsiella pneumoniae* from dairy cattle in northern Jiangsu, 208 dairy milk samples were collected from four dairy farms in northern Jiangsu. A total of 68 isolates were obtained through bacterial isolation, purification, and 16S rDNA identification. Eleven virulence genes were detected by specific PCR. The susceptibility of the isolates to antimicrobials was analyzed using the Kirby–Bauer method. The Pearson correlation coefficient was used to analyze the correlation between the presence of virulence genes and the phenotype of drug resistance. ST 2661 was the most prevalent type of *K. pneumoniae* (13/68, 19.1%) among the 23 ST types identified from the 68 isolates. The virulence gene *allS* was not detected, but the positive detection rates of the virulence genes *fimH*, *ureA*, *uge* and *wabG* were 100.0%. Notably, the detection rates of genes *rmpA* and *wcaG*, related to the capsular polysaccharide, were 4.4% and 11.8%, respectively, which were lower than those of genes related to siderophores (*kfuBC*, *ybtA* and *iucB* at 50.0%, 23.5%, and 52.9%, respectively). The *K. pneumoniae* isolates were sensitive to ciprofloxacin, nitrofurantoin, and meropenem. However, the resistance rate to penicillin was the highest (58/68, 85.3%), along with resistance to amoxicillin (16/68, 23.5%). The results revealed the distribution of 23 ST types of *K. pneumoniae* from the milk from bovine-mastitis-infected dairy cows in northern Jiangsu, and the expression or absence of the virulence gene *kfuBC* was related to the sensitivity to antibiotics. The current study provides important information relating to the distribution and characteristics of *K. pneumoniae* isolated from dairy cows with clinical bovine mastitis, and is indicative of strategies for improving the treatment of *K. pneumoniae*-induced bovine mastitis.

## 1. Introduction

Bovine mastitis is one of the major diseases threatening the development of the dairy industry, and it can lead to a decline in milk production and low quality of dairy products [1,2]. The reported incidence rate of bovine mastitis, including clinical and subclinical, in China is as high as 20–70%, and this results in huge economic losses [3]. Different pathogens, including bacteria, fungi, and viruses, as well as other types of physical damage, are usually regarded as factors causing mastitis worldwide [4]. Among the bacteria isolated from the milk of cows with clinical mastitis, *Klebsiella pneumoniae* (*K. pneumoniae*) is widely distributed in water, soil, and plants, and is a common Gram-negative environmental pathogen that causes mastitis [5,6]. Moreover, it has been suggested that infection with *K. pneumoniae* can induce pneumonia, bloodstream infection and pyogenic liver abscesses in mammals [5]. Hence, *K. pneumoniae* invasion is not only related to the infection of domestic animals, but also affects public health. Since the prevalence of the phenomenon of multidrug resistance (MDR) has increased, *K. pneumoniae* isolates have been found in many practical cases [6].

Currently, the use of antimicrobials is still the main therapeutic strategy for the treatment of bovine mastitis, and antimicrobials including β-lactams, quinolones, aminoglycosides, tetracyclines, and macrolides have been approved for application in livestock infections in the United States [7]. Notably, the prolonged abuse of antibiotics contributes to increased bacterial drug resistance and leads to difficulties associated with pathological bacterial infections [8]. For instance, quinolone-resistant *E. coli* has increased in China due to the overuse of quinolones in food-producing animals, such as dairy cows [9]. Moreover, isolated *E. coli* from some farms in China reveals resistance to penicillins, cephalosporins, and gentamycin with rates of 93–99%, 54–66%, and 82%, respectively [10]. Recently, *K. pneumoniae* isolated from the milk of both healthy and mastitis-infected dairy cows in China revealed a potential increase in drug resistance; 21.21% were resistant to tetracycline, 13.64% to chloramphenicol, and 12.12% to aminoglycosides [11]. Aside from Gram-negative bacteria, methicillin-resistant *Staphylococcus aureus* isolated from the milk of mastitis-infected dairy cows also raises great concern regarding the decreased efficacy of antimicrobials in Korea [12]. Hence, increased MDR in pathological bacteria allows pathogens to bypass destruction by antibiotics, and are able to persist and survive in the host.

In general, *K. pneumoniae* contains virulence factors, including fimbriae, capsular polysaccharides, lipopolysaccharides, and siderophores, which are encoded by *fimH*, *rmpA*, *maga*, *uge*, *wabg*, *iuc*, *iro*, and other virulence genes [13]. As reported, the expression of virulence genes is associated with the infectivity and pathogenicity of *K. pneumoniae* [14]. The dissemination of resistance is associated with genetic mobile elements, such as extended-spectrum β-lactamases (ESBL)-encoding plasmids that may also carry virulence factors [15]. A proficient pathogen is virulent, resistant to antibiotics, and epidemic. The interplay between resistance and virulence is poorly understood, and is receiving great attention [16]. The *K. pneumoniae* isolates that are resistant to cefoxitin, chloramphenicol, and quinolones have been found to have a decreased property of adhesion to Int-407 cells [17]. Moreover, the lower virulence in ESBL *K. pneumoniae* isolates compared to non-ESBL *K. pneumoniae* is mainly attributed to the lower carriage rate and higher mutation rate of rmpA-and *rmpA2*-encoded genes [14]. Nevertheless, the distribution of virulence gene expression and the correlation with MDR in *K. pneumoniae* isolates from the milk of cows with clinical mastitis remains unclear. In the present study, we used *K. pneumoniae* isolated from the milk of dairy cows with clinical mastitis from Jiangsu Province to investigate the distribution of virulence gene presence, and to test their antimicrobial susceptibility with defined MLST. This study provides insight into the spread of infection by *K. pneumoniae* and resistance in the case of bovine mastitis, as well as the potential target genes involved in the links between virulence determinants and MDR.

## 2. Materials and Methods

### 2.1. Isolation and Identification

Milk samples were collected from 208 mastitis-infected dairy cows between 2020 and 2021. The locations of the selected farms were distributed between different regions (Huaian, Yancheng, Sihong, and Yangzhou) in Jiangsu Province, China. All strains were isolated from the milk of dairy cattle with clinical signs of mastitis and tested using the California mastitis test (CMT), according to the recommendation of the U.S. National Mastitis Council [18,19]. A total of 100 μL of collected milk samples were plated onto blood agar containing 5% fresh sheep whole blood and incubated aerobically at 37 °C for 24 h. Based on the morphology of the colonies, a single identical colony from each sample was then sub-cultured by streaking on MacConkey (MC) agar. Plate-cultured bacteria were expanded aerobically in nutrient broth at 37 °C for 24 h. All of the suspected isolates were further confirmed using a DNA extraction kit (Tiangen, Beijing, China) and PCR assay (Vazyme, Nanjing, China) following 16S rDNA sequencing (Tsingke, Beijing, China) [20]. Purified PCR products were sequenced by pyrosequencing and sequences were proofread using SnapGene (GSL Biotech, Chicago, IL, USA). Then, the sequenced results from the PCR amplicons were analyzed via a BLAST sequence search using the NCBI database for species identification. The confirmed isolates were kept in 15% glycerol at −80 °C as frozen stock. *K. pneumoniae* isolates were characterized using multilocus sequence typing (MLST), antimicrobial susceptibility, and virulence gene expression.

### 2.2. MLST Analysis

MLST analyses of *K. pneumoniae* isolates were performed as described previously [21]. In brief, genomic DNA samples were extracted from the cultured bacterial strains using a TIANamp Bacteria DNA Kit (Tiangen, Beijing, China). Primers specific to 7 housekeeping genes (*rpoB*, *gapA*, *mdh*, *pgi*, *phoE*, *infB*, *and tonB*) were selected for PCR amplification (listed in Appendix A). PCR amplification was performed at an annealing temperature of 50 °C for all genes except for gapA (60 °C) and tonB (45 °C). Purified PCR products were sequenced by pyrosequencing, and sequences were proofread using SnapGene (GSL Biotech, Chicago, IL, USA). Allele numbers and sequence types were assigned using an online tool (https://bigsdb.pasteur.fr/klebsiella/, accessed on 6 March 2022). According to the ST results obtained from pubmlst, the alleles of the corresponding ST type in the pubmlst database are concatenated in the same order to obtain the concatenated sequences of different ST types. Then, sequence alignment is performed using the neighbor-joining algorithm, Bootstrap 1000, to obtain the phylogenetic tree as shown in Figure 1 [22].

### 2.3. Antimicrobial Susceptibility Test

Antimicrobial susceptibility testing was performed on Mueller–Hinton agar (MHA) using the Kirby–Bauer disk diffusion method [23]. The 11 commercial antibiotic disks included in the experiment included streptomycin (STR, 10 μg), ciprofloxacin (CIP, 5 μg), gentamycin (CN, 10 μg), meropenem (MEM, 10 μg), piperacillin (PRL, 100 μg), cephalothin (KF, 30 μg), penicillin (PEN, 10 μg), amoxicillin (AML, 20 μg), tetracycline (TCY, 30 μg), azithromycin (AZM, 15 μg), and nitrofurantoin (NIT, 300 μg). The antimicrobial disks were obtained from Microbial Reagent (Hangzhou Microbial Reagent Co., Hangzhou, China). The *E. coli* ATCC 25922 strain was used as the control strain, and the test results were then recorded as susceptible or resistant according to the zone diameter interpretative standards of the Clinical and Laboratory Standards Institute [24], and when this was not available, according to the antimicrobial disk manufacturer’s instructions.

### 2.4. Virulence Gene Detection

The presence of 11 virulence genes carried by *K. pneumoniae* was detected by performing PCR assays in 68 isolated *K. pneumoniae* isolates in the current study, including *rmpA*, *wcaG*, *allS*, *kfuBC*, *ybtA*, *iucB*, *iroNB*, *fimH*, *ureA*, *uge*, and *wabG* [25,26]. The PCR was performed as follows: denaturing at 94 °C for 3 min, then 35 cycles at 94 °C for 40 s, 55 °C for 40 s, and 72 °C for 1 min, along with a final elongation at 72 °C for 10 min. ATCC 700603 and ATCC 25922 were taken as the reference strains for PCRs. Amplified PCR products were then verified with 1% agarose gel and DNA sequencing (Tsingke, Beijing, China). The presence of the targeted genes was identified by a visible band on the gel images.

### 2.5. Statistical Analysis

The Pearson correlation coefficient, as a statistical index reflecting the degree of correlation between variables, was set between −1 and 1. SPSS 26.0 software was used to analyze the Pearson correlation coefficient between virulence genes and drug resistance phenotypes (IBM Inc., Armonk, NY, USA).

## 3. Results

### 3.1. Microbiological Characterization and Sequence Types of K. pneumoniae Isolates

*K. pneumoniae* were isolated from the 208 milk samples from mastitis-infected dairy cows between 2020 and 2021 on farms located in different regions of Jiangsu Province, China. In the present study, the *K. pneumoniae* isolates were identified by 16S rRNA analysis, 68 in total, with 15 isolates on Farm A, 21 isolates on Farm B, 8 isolates on Farm C, and 24 isolates on Farm D. The BLAST program in NCBI displayed a high similarity (>99%) to those of *K. pneumoniae* subspecies.

In this study, a total of 23 known STs were found from the 68 *K. pneumoniae* isolates. As listed in Table 1, of the 23 STs, ST 2661 was identified with 13 isolates (19.1%), whereas ST 43 and ST 2410 were identified with 7 isolates (10.3%) and 6 isolates (8.8%), respectively. The phylogenetic relationship of 23 STs, based on the concatenated sequences of seven MLST alleles and the neighbor-joining method, resulted in two clusters (Figure 1). Cluster I was mainly sourced from bovines, while Cluster II was sourced from humans, according to the dataset from Institut Pasteur (https://bigsdb.pasteur.fr/klebsiella/, accessed on 6 March 2022).

### 3.2. Antimicrobial Susceptibility Determination

We further investigated the antibiotic resistance profiles of *K. pneumoniae* isolates. The antimicrobial susceptibilities of 68 *K. pneumoniae* isolates are shown in Figure 2. In general, isolates of *K. pneumoniae* were resistant to penicillin and amoxicillin, at 85.3% and 67.6%, respectively. Moreover, β-lactams, including cephalothin and piperacillin, displayed similar levels of resistance, with 11.8% and 10.3% resistance by *K. pneumoniae* isolates, respectively. Notably, among the isolates isolated in the present study, a rate of 41.2% was found in terms of the resistance to streptomycin, which is relatively higher than that for gentamycin (5.9%). Additionally, 17.6% of the *K. pneumoniae* (12/68) isolates were resistant to tetracycline. In contrast, all of the 68 *K. pneumoniae* isolates were susceptible to meropenem, nitrofurantoin, and ciprofloxacin. Thirteen *K. pneumoniae* isolates (19.1%) showed acquired resistance to more than three classes of antimicrobials, including tetracyclines, aminoglycosides, and β-lactams. Among the MDR *K. pneumoniae* isolates, two isolates displayed resistance to seven antibiotics. Additionally, only one isolate of *K. pneumoniae* showed no resistance to the tested antibiotics in the current study.

### 3.3. Virulence Gene Detection

In the present study, we selected 11 genes in the presence of the isolated *K. pneumoniae* isolates, including allantoins (*allS*), pilin (*fimH*), lipopolysaccharides (*ureA*, *uge*, *wabG*), capsular polysaccharides (*rmpA*, *wcaG*), and siderophores (*kfuBC*, *ybtA*, *iucB*, *iroNB*), related to bacterial virulence. The data are presented in Table 2. Of the 11 investigated virulence genes, the *fimH*, *urea*, *uge*, and *wabG* genes were detectable in all 68 isolates (100%), whereas *allS* was not detected in any of the isolates (0%). Six isolates harbored at least four virulence genes (8.8%), while four isolates were detected for, at most, eight virulence genes (5.9%). Among the *K. pneumoniae* isolates, 36 isolates carried by 5 virulence genes were detected most often (52.9%), whereas 17 isolates were simultaneously positive to *kfuBC*, *fimH*, *ureA*, *uge*, and *wabG*, and 15 isolates were simultaneously positive to *iucB*, *fimH*, *urea*, *uge*, and *wabG*. Notably, siderophore-related virulence genes were detected more often than the genes related to capsular polysaccharides, pilin, or lipopolysaccharides.

### 3.4. Correlations between Virulence Gene Frequency and Antimicrobial Susceptibility

In order to uncover the interaction between the presence of virulence genes and antimicrobial resistance, we analyzed the correlation of the frequency of each virulence gene with antimicrobial susceptibility in this study. As the results show in Table 3, there was no significant correlation between the number of virulence genes carried by *K. pneumoniae* isolates and the incidence of phenotypic antibiotic resistance. However, the carrier rate of the *kfuBC* gene was negatively correlated with the incidence of *K. pneumoniae* drug resistance (−0.348). The *kfuBC* gene carried by *K. pneumoniae* isolates was negatively correlated with resistance to tetracycline (−0.463), piperacillin (−0.242), and streptomycin (−0.299). Moreover, the detection rates of the *rmpA* and *iroNB* genes carried by *K. pneumoniae* isolates were significantly positively correlated with resistance to tetracycline (0.276) and streptomycin (0.257). Furthermore, *iucB*-positive isolates displayed a significantly positive correlation with tetracycline (0.359) and streptomycin (0.370).

## 4. Discussion

Since the climate in this area is characterized by a higher temperature and humidity than other areas in the northern part of China, pathogenic bacterial infection is particularly troublesome, especially in dairy farming. *K. pneumoniae* has traditionally been considered an opportunistic pathogen, and has recently been regarded as one of the main pathogenic bacteria resulting in severe bovine mastitis [11,27]. The *K. pneumoniae* epidemic poses a great challenge to the dairy farming industry, seriously affecting the economic efficiency of large-scale farms and threatening the health of both herds and humans [28,29]. Significant differences in the prevalence of *K. pneumoniae* in different regions have been reported; therefore, it is crucial to further investigate the prevalence of *K. pneumoniae* on farms in northern Jiangsu Province. The experiment conducted MLST typing on 68 *K. pneumoniae* isolates isolated from 208 milk samples which were collected from four farms in northern Jiangsu Province. A total of 23 sequence types were detected, reflecting the genetic diversity of *K. pneumoniae*. Among them, ST2661, ST43, and ST2410 were detected at higher rates of 19.1% (13/68), 10.3% (7/68), and 8.8% (6/68), respectively, indicating that these three ST types of *K. pneumoniae* were the dominant sequence types in the tested collection of farms in northern Jiangsu. However, the carbapenem-resistant *K. pneumoniae* ST11, a widely prevalent type in China, as reported previously, was not detected in the present study [30]. Notably, the fact is that the prevalent ST11 in China has been mainly identified in humans and is carbapenem resistant, and its prevalence in dairy farms is not well studied. The detection of two strains belonging to ST37 was consistent with the finding that ST37 was present in humans, chickens, and cattle, as reported previously [31,32]. MLST typing of *K. pneumoniae* isolates in this study helped to further analyze the phylogeny and evolution of different subtypes of *K. pneumoniae* which are prevalent in northern Jiangsu.

Antibiotic therapy has become an important strategy in the fight against bacterial infections. However, prolonged or irregular use of antibiotics has also caused increased bacterial resistance and even the emergence of multidrug-resistant bacteria [33]. The results showed that *K. pneumoniae* isolates maintained different phenotypes of resistance to tetracyclines, aminoglycosides, and β-lactams in the present study. All *K. pneumoniae* isolates were completely susceptible to ciprofloxacin, nitrofurantoin, and meropenem, with a resistance rate of 0.0%, which is consistent with the reported *K. pneumoniae* resistance in the Jiangsu (65 isolates) and Shandong (1 isolate) regions [11]. This might be related to the rare use of ciprofloxacin, nitrofurantoin, and meropenem on the sampled farms. In contrast, *K. pneumoniae* isolates demonstrated severe resistance to penicillin and amoxicillin, with resistance rates of 85.3% and 67.6%, respectively, indicating that the *K. pneumoniae* isolated from the farms in northern Jiangsu Province were more frequently exposed to β-lactam antibacterial drugs, probably due to the more widespread use of β-lactam antibacterial drugs on the farms in the region. Thirteen isolates of MDR *K. pneumoniae* were detected, with five different resistance profiles, and most of the *K. pneumoniae* isolates (7/68, 10.3%) were simultaneously resistant to tetracycline, streptomycin, penicillin, and amoxicillin, which was inconsistent with the results reported by Rafael et al. [34]. This might be attributed to the differences in the geographical locations of *K. pneumoniae*, frequency of use of the antimicrobial reagents, and the transfer efficiency of resistant genes between the two studies.

*K. pneumoniae* carries virulence genes whose encoded proteins are involved in the processes of invasion, colonization, adhesion, and resistance to phagocytosis, which is associated with the pathogenicity of *K. pneumoniae* [16]. Eleven virulence factor-related genes were selected in this study, of which ten virulence genes were detectable, with the exception of the *allS* gene that is associated with allantoin. Among them, 100.0% of the detection rate was for the *fimH*, *ureA*, *uge*, and *wabG* genes. The *fimH* gene is one of the type I pilus-related virulence genes, and its encoded fimH adhesin is an important mediator of *K. pneumoniae* adhesion to target cells. It has been reported that type I pili are expressed in 90% of *K. pneumoniae*, and play an important role in escaping host immune action and biofilm formation [35]. In this study, *ureA*, *uge*, and *wabG*, associated with lipopolysaccharide formation, were present in all *K. pneumoniae* isolates, which is consistent with previous reports that most *K. pneumoniae* carry *uge* genes and *wabG* genes at rates of 88% to 100% [36]. This is indicative of the prevalent virulence genes in northern Jiangsu being conserved due to the structure necessary for Gram-negative bacteria. The virulence gene *rmpA*, related to capsular polysaccharides, was mostly expressed in the plasmids of high-virulence *K. pneumoniae* (hv *K. pneumoniae*). Its expression with *wcaG* could enhance the ability of *K. pneumoniae* to resist neutrophil phagocytosis [37]. However, the current study detected a low positive presence of *rmpA* and *wcaG* (4.4% and 11.8%, respectively), indicating that *K. pneumoniae* virulence in this region was not solely regulated by capsular polysaccharides. Siderophores are one of the main virulence factors of *K. pneumoniae*, which allow *K. pneumoniae* to chelate iron from the environment and aid in its growth [38]. Siderophores can be classified into enterobacteriaceae, aerobacteriaceae, yersiniaceae, and salmonellaceae. The *iucB* gene is a type of aerobacteriaceae virulence gene, with the highest detection rate (52.9%) among the four siderophore virulence genes selected for the experiment, followed by the *kfuBC* virulence gene, with a higher detection rate of 50.0%. This indicates that the *iucB* and *kfuBC* genes are more frequently expressed in *K. pneumoniae* siderophores in northern Jiangsu Province. The *K. pneumoniae* isolates carried between four and eight virulence genes, at least, and the number of *K. pneumoniae* isolates carrying five virulence genes was the highest detected. The results suggested that *K. pneumoniae* in northern Jiangsu carried multiple virulence genes, and varied by distribution characteristics and origin, which differed from the results reported previously [39].

To our knowledge, there are relatively few studies on the relationship between virulence genes carried by *K. pneumoniae* and its drug resistance [16]. In this study, the correlation between virulence genes and drug resistance of *K. pneumoniae* isolates was analyzed, and the results showed that the number of virulence genes carried by *K. pneumoniae* isolates was not significantly correlated with the type of drug resistance. This was consistent with the results reported by Zhao et al. [40]. However, Wu et al. reported a significant negative correlation between the number of virulence genes expressed and the incidence of drug resistance, which might explain the differences in biological characteristics, infection routes, and environmental changes between *K. pneumoniae* isolates of bovine and human origin [41]. Moreover, the types and strengths of antimicrobial drugs used as treatments are the predominant factors for drug resistance development. The presence of the virulence genes *rmpA*, *kfuBC*, *iucB*, and *iroNB* was correlated with drug resistance in this study. The capsular-polysaccharide-related gene *rmpA* was significantly and positively correlated with resistance to tetracycline (0.276) and streptomycin (0.257), which was inconsistent with the results of the correlation between *rmpA* and the drug susceptibility phenotype reported by Wang et al. [42]. Generally, the presence of any components of siderophore clusters in *K. pneumoniae* isolates suggests the risk of serious infection [43]. For example, the ferric-uptake operon-related gene *kfuBC* was suggested to be highly associated with tissue invasion and clinical mastitis cases in dairy cows [39,40]. Notably, the highly positive rate of *kfuBC* detected in *K. pneumoniae* isolates was significantly negatively correlated with its corresponding resistance characteristics to antibiotics, revealing its role as a key gene for *K. pneumoniae* pathogenicity that effectively regulates multidrug resistance in pathogenic bacteria. The simultaneous presence of virulence and resistance factors in one isolate could be explained by the presence of mosaic plasmids carrying both virulence and resistance genes, which is not uncommon for *K. pneumoniae* [44,45,46]. However, further studies are needed for the verification of the interplay between the emergence of multidrug-resistance and virulence regulators in *K. pneumoniae* isolates, as well as the investigation of plasmid replicons for virulence and resistance genes. Moreover, it is expected that the key molecular targets that regulate *K. pneumoniae* virulence and drug resistance will be explored in a further investigation.

## 5. Conclusions

Collectively, in this study, ST 2661 was the most common of the 23 isolated MLST types in northern Jiangsu Province. The emergence of multidrug resistance in bacteria isolated from the milk of cows with clinical mastitis indicates the increasing difficulties of treatment for mammary infections using antibiotics. Moreover, the results of the correlation between siderophore-secreting-related *kfuBC* and antimicrobial susceptibility underscored the potential links between virulence and multidrug resistance. Our study provides important information for the distribution and characteristics of pathogenic *K. pneumoniae*-infected bovine mastitis and the improvement of antimicrobial selection for the treatment.

## Figures and Tables

**Figure 1 animals-12-02627-f001:**
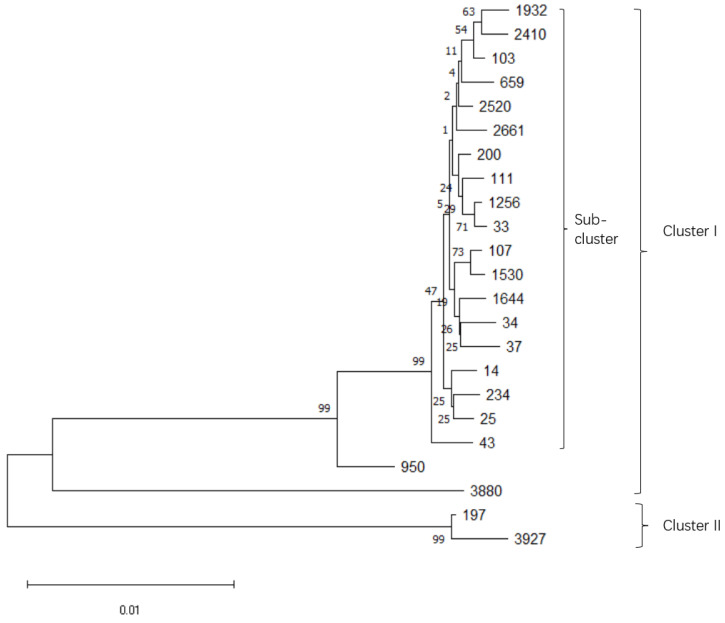
Phylogenetic organization of *K. pneumoniae* isolates of different MLST based on the neighbor-joining method.

**Figure 2 animals-12-02627-f002:**
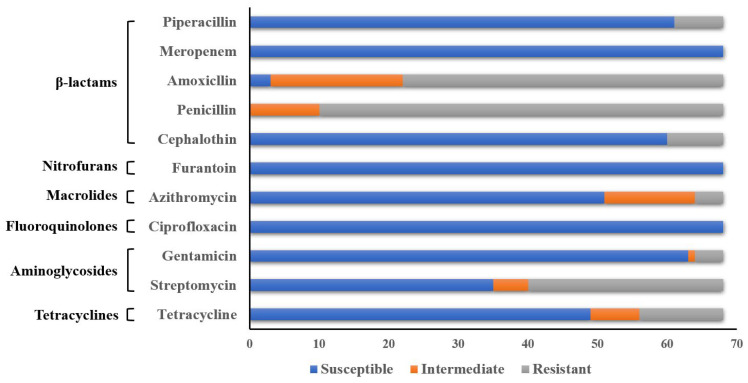
Proportions of drug-resistant phenotypes of *K. pneumoniae.* All of the 68 isolates of *K. pneumoniae* were analyzed with six groups of antimicrobials of 11 antimicrobial agents. The bars in blue, red, and gray designate antimicrobial susceptibility as susceptible, intermediate, and resistant, respectively.

**Table 1 animals-12-02627-t001:** Analysis of MLST for 68 *K. pneumoniae* isolates.

ST	MLST Allele	Isolates (No.)	Proportion (%)
ST14	1-6-1-1-1-1-1	4	5.9
ST25	2-1-1-1-10-4-13	3	4.4
ST33	2-3-5-1-12-4-9	1	1.5
ST34	2-3-6-1-9-7-4	2	2.9
ST37	2-9-2-1-13-1-16	2	2.9
ST43	2-6-1-5-11-1-15	7	10.3
ST103	4-1-1-1-9-1-6	1	1.5
ST107	2-1-2-17-27-1-39	3	4.4
ST111	2-1-5-1-17-4-42	1	1.5
ST197	16-28-21-27-47-22-67	1	1.5
ST200	2-1-2-1-12-1-68	1	1.5
ST234	2-1-2-1-7-1-24	3	4.4
ST659	66-1-65-1-9-11-8	3	4.4
ST950	2-1-1-20-56-4-31	3	4.4
ST1256	2-1-58-1-12-4-220	4	5.9
ST1530	1-1-1-3-27-1-39	3	4.4
ST1644	2-95-2-3-1-4-274	1	1.5
ST1932	10-7-1-26-10-4-127	1	1.5
ST2410	4-1-13-2-10-44-6	6	8.8
ST2520	2-1-2-1-9-1-42	3	4.4
ST2661	4-7-2-1-9-4-25	13	19.1
ST3880	18-22-26-22-154-13-165	1	1.5
ST3927	16-24-21-138-153-40-67	1	1.5

**Table 2 animals-12-02627-t002:** The detection of 11 selected virulence genes related to capsular polysaccharides, allantoins, siderophores, pilis, and lipopolysaccharides.

Types	Virulence Genes	Isolates (No.)	Positive Rates (%)
Capsular polysaccharides	rmpA	3	4.4
	wcaG	8	11.8
Allantoins	allS	0	0.0
Siderophores	kfuBC	34	50.0
	ybtA	16	23.5
	iucB	36	52.9
	iroNB	3	4.4
Pilis	fimH	68	100.0
Lipopolysaccharides	ureA	68	100.0
	uge	68	100.0
	wabG	68	100.0

**Table 3 animals-12-02627-t003:** The correlation analysis between the presence of virulence genes and phenotypes of drug resistance.

Antimicrobial	Virulence Genes						
	No.	rmpA	wcaG	kfuBC	ybtA	iucB	iroNB
R (no.)	0.015	0.176	−0.143	−0.348 **	0.110	0.231	0.176
TE	-	0.276 *	−0.049	−0.463 **	−0.075	0.359 **	0.276 *
S	-	0.257 *	−0.213	−0.299 *	0.099	0.370 **	0.257 *
GM	-	−0.054	−0.091	−0.125	−0.139	−0.015	−0.054
AZM	-	−0.054	−0.091	0.025 *	−0.139	−0.265 *	−0.054
CEF	-	−0.078	0.008	−0.183	0.120	−0.113	−0.078
P	-	0.089	0.023	0.000	0.132	0.191	0.089
AMX	-	0.149	−0.040	−0.189	0.161	0.167	0.149
PIP	-	−0.073	−0.124	−0.242 *	0.154	−0.068	−0.073
βL	-	0.046	0.078	−0.072	0.119	0.228	0.046

** indicates *p* < 0.01; * indicates *p* < 0.05; TE, Tetracycline; S, Streptomycin; GM, Gentamycin; AZM, Azithromycin; CEF, Cephalothin; P, Penicillin; AMX, Amoxicillin; PIP, Piperacillin; βL, β-lactams.

## Data Availability

The datasets used and/or analyzed during the current study are available from the corresponding author upon reasonable request.

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
