# Peer review of "The Characteristics of Multilocus Sequence Typing, Virulence Genes and Drug Resistance of Klebsiella pneumoniae Isolated from Cattle in Northern Jiangsu, China"

_animals, 2022, doi:10.3390/ani12192627_

Round 1
Reviewer 1 Report
Authors characterize K. pneumoniae isolates recovered from the milk of dairy cows with clinical mastitis from Jiangsu Province, determining the presence of virulence genes, the antimicrobial susceptibility profile and the genetic relationship by multilocus sequence typing (MLST) analysis. The subject is relevant and it should be of interest for Journal's readers. However, a minor revision is necessary. I am listing below points that should be addressed by Authors.
“Currently, the use of antimicrobials is still the main therapeutic strategy for the treatment of bovine mastitis, and antimicrobials including β-lactams, quinolones, aminoglycosides, tetracyclines and macrolides have been approved for application in livestock infections in the United States.”
> Provide reference
“Notably, the prolonged abuse of antibiotics contributes to increased bacterial drug resistance and leads to the difficulties associated with pathological bacterial infections.”
> Provide reference
“For instance, quinolone-resistant E. coli have increased in China due to the overuse of quinolones in food-producing animals, such as dairy cows[9]. Moreover, isolated E. coli strains from some farms in China reveal resistance to penicillins, cephalosporins and gentamycin with rates of 93-99%, 54-66% and 82%, respectively [10].”
> Authors should highlighted K. pneumoniae.
“K. pneumoniae mainly contains virulence factors, including fimbriae, capsular polysaccharides, lipopolysaccharides and siderophores, that are encoded by FimH, rmpA, Maga, uge, wabg, IUC, iro and other virulence genes.”
> Provide reference
“As reported, the expression of virulence genes is associated with the infectivity and pathogenicity of K. pneumoniae.”
>Provide references
“The dissemination of resistance is associated with genetic mobile elements, such as ESBL-encoding plasmids that may also carry virulence factors. A proficient pathogen is virulent, resistant to antibiotics and epidemic. The interplay between resistance and virulence is poorly understood and is receiving great attention.”
> Provide references
“The dissemination of resistance is associated with genetic mobile elements, such as ESBL-encoding plasmids”
> Authors should define ESBL.
“Province to investigate the distribution of virulence gene expression and test their antimicrobial susceptibility with defined MLST.”
> Virulence gene expressions were not investigated. This paper evaluates the presence of virulence genes. The only presence of these genes not mean expression necessarily.
> Bacterial identification is not clear .
(Bacterial identification, MLST)
>Details about DNA sequencing are missing.
(listed in Table 1). Table 1 is not necessary.
“the test results were then recorded as susceptible or resistant based on the zone diameter interpretative standards of the Clinical and Laboratory Standards Institute [20] and, when not available, according to the antimicrobial disk manufacturer's instructions.”
[20] Wayne, P.A. Clinical and laboratory standards institute. Performance standards for antimicrobial susceptibility testing. 2011.
> Authors need to use a recent CLSI version (CLSI 2021/CLSI 2022).
“The presence of 11 virulence genes carried by K. pneumoniae was detected by per- 134
forming PCR assays in 68 isolated K. pneumoniae strains in the current study, including 135
rmpA, wcaG, allS, kfuBC, ybtA, iucB, iroNB, fimH, ureA, uge and wabG.”
> Provide reference about PCR primers
Strains of K. pneumoniae were isolated from the 208 milk samples from mastitis-infected dairy cows during 2020-2021 on farms located in different regions in Jiangsu Province, China. Since the climate in this area is characterized by a higher temperature and
humidity than other areas in the northern part of China, pathogenic bacterial infection is particularly troublesome, especially in dairy farming.
> These sentences/data are not results. Authors should revise.
Figure 1. Phylogenetic organization of K. pneumoniae isolates of different MLST based on neighbor joining method.
How did authors get to figure 1?
“The phylogenetic relationship of 23 STs, based on the concatenated sequences of seven MLST alleles and the neighbor-joining method, resulted in two clusters (Figure 2).
> Figure 2 displayed Antimicrobial susceptibility results.
“Our study provides important information for the development of strategies for managing bovine mastitis and improving antimicrobial selection, especially for pathogenic K. pneumoniae infections.”
> Did the results support this conclusion? Authors should revise conclusions.
“Virulence gene detection”
> Author should provide a table about co-existence of virulence genes.
Author Response
Dear Editors and Reviewers:
Thank you for your letter and for the reviewers’ comments concerning our manuscript entitled “The characteristics of multilocus sequence typing, virulence genes and drug resistance of Klebsiella pneumoniae isolated from cattle in northern Jiangsu, China”. Those comments are all valuable and very helpful for revising and improving our paper, as well as the important guiding significance to our researches. We have studied comments carefully and have made correction which we hope meet with approval. Revised portion are highlighted in yellow in the paper. The main corrections in the paper and the responds to the reviewer’s comments are as flowing:
Reviewer 1
Authors characterize K. pneumoniae isolates recovered from the milk of dairy cows with clinical mastitis from Jiangsu Province, determining the presence of virulence genes, the antimicrobial susceptibility profile and the genetic relationship by multilocus sequence typing (MLST) analysis. The subject is relevant and it should be of interest for Journal's readers. However, a minor revision is necessary. I am listing below points that should be addressed by Authors.
“Currently, the use of antimicrobials is still the main therapeutic strategy for the treatment of bovine mastitis, and antimicrobials including β-lactams, quinolones, aminoglycosides, tetracyclines and macrolides have been approved for application in livestock infections in the United States.”
> Provide reference
RE: The reference has been added to the revised manuscript.
“Notably, the prolonged abuse of antibiotics contributes to increased bacterial drug resistance and leads to the difficulties associated with pathological bacterial infections.”
> Provide reference
RE: The reference has been added to the revised manuscript.
“For instance, quinolone-resistant E. coli have increased in China due to the overuse of quinolones in food-producing animals, such as dairy cows[9]. Moreover, isolated E. coli strains from some farms in China reveal resistance to penicillins, cephalosporins and gentamycin with rates of 93-99%, 54-66% and 82%, respectively [10].”
> Authors should highlighted K. pneumoniae.
RE: Thanks for the suggestion, the evidence for KP has been described following the E.coli description.
“K. pneumoniae mainly contains virulence factors, including fimbriae, capsular polysaccharides, lipopolysaccharides and siderophores, that are encoded by FimH, rmpA, Maga, uge, wabg, IUC, iro and other virulence genes.”
> Provide reference
RE: The reference has been added to the revised manuscript.
“As reported, the expression of virulence genes is associated with the infectivity and pathogenicity of K. pneumoniae.”
>Provide references
RE: The reference has been added to the revised manuscript.
“The dissemination of resistance is associated with genetic mobile elements, such as ESBL-encoding plasmids that may also carry virulence factors. A proficient pathogen is virulent, resistant to antibiotics and epidemic. The interplay between resistance and virulence is poorly understood and is receiving great attention.”
> Provide references
RE: The reference has been added to the revised manuscript.
“The dissemination of resistance is associated with genetic mobile elements, such as ESBL-encoding plasmids”
> Authors should define ESBL.
RE: The ESBL has been defined in the revised manuscript.
“Province to investigate the distribution of virulence gene expression and test their antimicrobial susceptibility with defined MLST.”
> Virulence gene expressions were not investigated. This paper evaluates the presence of virulence genes. The only presence of these genes not mean expression necessarily.
RE: Thanks for the suggestion, the word has been replaced in the revised manuscript.
> Bacterial identification is not clear .
RE: The detailed information has been added to the revised manuscript.
(Bacterial identification, MLST)
>Details about DNA sequencing are missing.
RE: Thanks for the suggestion, the detailed information for DNA sequencing has been added to the revised manuscript.
(listed in Table 1). Table 1 is not necessary.
RE: Thanks for the suggestion, we have put the table to supplementary file.
“the test results were then recorded as susceptible or resistant based on the zone diameter interpretative standards of the Clinical and Laboratory Standards Institute [20] and, when not available, according to the antimicrobial disk manufacturer's instructions.”
[20] Wayne, P.A. Clinical and laboratory standards institute. Performance standards for antimicrobial susceptibility testing. 2011.
> Authors need to use a recent CLSI version (CLSI 2021/CLSI 2022).
RE: Thanks for the suggestion. This has been modified and updated in the revised manuscript.
“The presence of 11 virulence genes carried by K. pneumoniae was detected by per- 134
forming PCR assays in 68 isolated K. pneumoniae strains in the current study, including 135
rmpA, wcaG, allS, kfuBC, ybtA, iucB, iroNB, fimH, ureA, uge and wabG.”
> Provide reference about PCR primers
RE: The references have been added to the revised manuscript.
Strains of K. pneumoniae were isolated from the 208 milk samples from mastitis-infected dairy cows during 2020-2021 on farms located in different regions in Jiangsu Province, China. Since the climate in this area is characterized by a higher temperature and
humidity than other areas in the northern part of China, pathogenic bacterial infection is particularly troublesome, especially in dairy farming.
> These sentences/data are not results. Authors should revise.
RE: Thanks for the suggestion. We have moved the sentences to the first paragraph of Discussion section in the revised manuscript.
Figure 1. Phylogenetic organization of K. pneumoniae isolates of different MLST based on neighbor joining method.
How did authors get to figure 1?
RE: According to the ST results obtained from pubmlst, the alleles of the corresponding ST type in the pubmlst database are concatenated in the same order to obtain the concatenated sequences of different ST types, and then sequence alignment is performed using the neighbor joining algorithm, Bootstrap 1000, to obtain the phylogenetic figure. This information has been added to the revised manuscript.
“The phylogenetic relationship of 23 STs, based on the concatenated sequences of seven MLST alleles and the neighbor-joining method, resulted in two clusters (Figure 2).
> Figure 2 displayed Antimicrobial susceptibility results.
RE: The misnumber has been fixed in the revised manuscript.
“Our study provides important information for the development of strategies for managing bovine mastitis and improving antimicrobial selection, especially for pathogenic K. pneumoniae infections.”
> Did the results support this conclusion? Authors should revise conclusions.
RE: The conclusions have been modified as suggested in the revised manuscript.

Reviewer 2 Report
Xu et al. describe the molecular analysis of Klebsiella pneumoniae isolates obtained from cattle in China. In general, the manuscript is well written and easy to comprehend. Unfortunately, I found that the study design and main results including the correlation of virulence and resistance genes are questionable. I have several comments to be addressed.
Major comments:
- Most correlation results in the study were not significant. The simultaneous presence of virulence and resistance factors in one isolate could be explained by the presence of mosaic plasmid carrying both virulence and resistance genes, which is not uncommon for K. pneumoniae. The presence of plasmid replicons were not investigated, and thus the results of correlation studies cannot be used for deriving the reliable conclusions. I suggest performing additional analysis or completely remove the correlation part from the manuscript.
- Line 160 – “based on the concatenated sequences of seven MLST alleles” – Simple concatenation of the sequences is not an appropriate method for cluster analysis. Please describe which software you used for building the tree or give the reference with method description. Usually, eBURST algorithm is used for clustering the isolates based on MLST, but it relies on MLST allele differences, not the sequences of alleles themselves.
Minor comments:
- I suggest adding “Jiangsu, China” to the title since readers from other parts of the world may not be familiar with China provinces.
- The abstract almost twice exceeds the recommended length and contains plenty of non-essential data. I suggest removing the part starting from “The positive detection of the virulence…” and ending with “piperacillin (r=-0.242)”. The abstract should contain the essential results and brief conclusions only, while the data given above should better fit the Results section
- I suggest removing “The ST types of the isolates were identified by MLST analysis”. ST types can only be identified by MLST analysis – either by PCR on in silico. Other types (cgMLST, KL/O) are not called ST. MLST was already mentioned above
- I also suggest removing the phrase “it is expected that the key molecular targets…” since future perspectives better fit the Discussion section
Line 46 – the incidence of 20-70% is a very rough estimate. Are there some median or average values available for some regions of China or country in general? Are the data available for Jiangsu province?
Line 73 – “K. pneumoniae mainly contains virulence factors” – in fact, K. pneumoniae can contain many more virulence factors, e.g., mrk, sil, rmpA etc, which are rathe common in clinical and environmental population. Please rephrase.
Line 74-75 – the genes should be fimH and iuc (small letters)
Line 77 – please define ESBL here
Line 84 – please italicize gene names. Also check this throughout the manuscript (lines 115, 117, 136, 191 etc)
Line 109 – “K. pneumonias” – please correct a typo
Table 1 – I suggest moving the table to supplementary materials, if the primer sequences were not obtained by authors and were derived from the literature
Line 161 – figure 2 does not contain phylogenetic tree – it appears in the figure 1. In addition, the figures are not numbered in the order they appear in the manuscript. Probably, the numbers should be swapped. Please fix. The same is true for line 171 (should be figure 2, not 1).
Table 2 – MLST allele information is not necessary – it is common knowledge. The reference to bigSDB database will be enough
Table 2 – the STs are not sorted numerically. E.g., ST25 should come before ST107 – this will make the table more human-readable.
Lines 171-180 and throughout the manuscript – I think that ‘isolates’ would be more appropriate than ‘strains’ in this case because strain definition usually is more specific than the point of isolation. These terms are not synonymous
Line 236 – why the widely prevalent ST was not detected in your study? This fact should be discussed further
Line 263 – what was the reason for selection of these particular virulence genes? E.g., genes from mrk cluster or heave metal-resistance genes are also important
Line 271 – “were completely expressed” – you have not confirmed the gene expression – only its presence. Please fix.
Author Response
Dear Editors and Reviewers:
Thank you for your letter and for the reviewers’ comments concerning our manuscript entitled “The characteristics of multilocus sequence typing, virulence genes and drug resistance of Klebsiella pneumoniae isolated from cattle in northern Jiangsu, China”. Those comments are all valuable and very helpful for revising and improving our paper, as well as the important guiding significance to our researches. We have studied comments carefully and have made correction which we hope meet with approval. Revised portion are highlighted in yellow in the paper. The main corrections in the paper and the responds to the reviewer’s comments are as flowing:
Reviewer 2
Xu et al. describe the molecular analysis of Klebsiella pneumoniae isolates obtained from cattle in China. In general, the manuscript is well written and easy to comprehend. Unfortunately, I found that the study design and main results including the correlation of virulence and resistance genes are questionable. I have several comments to be addressed.
Major comments:
- Most correlation results in the study were not significant. The simultaneous presence of virulence and resistance factors in one isolate could be explained by the presence of mosaic plasmid carrying both virulence and resistance genes, which is not uncommon for K. pneumoniae. The presence of plasmid replicons were not investigated, and thus the results of correlation studies cannot be used for deriving the reliable conclusions. I suggest performing additional analysis or completely remove the correlation part from the manuscript.
RE: Thanks for the comments on this. We agree with the concerns from the reviewer that plasmid carrying both resistance and virulence genes contribute to the interactions of virulence and resistance in isolates. However, the correlation of virulence genes and phenotype of drug resistance may differ from the simultaneous presence of the virulence and resistance genes in the plasmid, to some extent, the chromosomal located resistance gene also results in the phenotypic resistance. In this study, we provide potential links in between virulence genes and phenotype of resistance, whereas the resistance genes yet not involved. Further study is certainly needed to uncover the relationship in these. However, we have added the discussion as suggested to the Discussion section in the revised manuscript.
- Line 160 – “based on the concatenated sequences of seven MLST alleles” – Simple concatenation of the sequences is not an appropriate method for cluster analysis. Please describe which software you used for building the tree or give the reference with method description. Usually, eBURST algorithm is used for clustering the isolates based on MLST, but it relies on MLST allele differences, not the sequences of alleles themselves.
RE: According to the ST results obtained from pubmlst, the alleles of the corresponding ST type in the pubmlst database are concatenated in the same order to obtain the concatenated sequences of different ST types, and then sequence alignment is performed using the neighbor joining algorithm, Bootstrap 1000, to obtain the phylogenetic figure. This information has been added to the revised manuscript.
Minor comments:
- I suggest adding “Jiangsu, China” to the title since readers from other parts of the world may not be familiar with China provinces.
RE: Thanks for the suggestion, we have made correction to the revised manuscript.
- The abstract almost twice exceeds the recommended length and contains plenty of non-essential data. I suggest removing the part starting from “The positive detection of the virulence…” and ending with “piperacillin (r=-0.242)”. The abstract should contain the essential results and brief conclusions only, while the data given above should better fit the Results section
RE: Thanks for the suggestion, we have shortened the size of the abstract section as recommended in the revised manuscript.
- I suggest removing “The ST types of the isolates were identified by MLST analysis”. ST types can only be identified by MLST analysis – either by PCR on in silico. Other types (cgMLST, KL/O) are not called ST. MLST was already mentioned above
RE: This sentence has been removed as recommended.
- I also suggest removing the phrase “it is expected that the key molecular targets…” since future perspectives better fit the Discussion section
RE: The sentence has been moved to the Discussion section in the revised manuscript.
Line 46 – the incidence of 20-70% is a very rough estimate. Are there some median or average values available for some regions of China or country in general? Are the data available for Jiangsu province?
RE: The scale of the rate covered both clinical and subclinical incidence of bovine mastitis. Since the clinical one is less diagnosed than that in the subclinical one. Hence, the scale looks a rough estimate. We have modified the description in the revised manuscript.
Line 73 – “K. pneumoniae mainly contains virulence factors” – in fact, K. pneumoniae can contain many more virulence factors, e.g., mrk, sil, rmpA etc, which are rathe common in clinical and environmental population. Please rephrase.
RE: Thanks for the suggestions, we have rephrased the word in the revised manuscript as recommended.
Line 74-75 – the genes should be fimH and iuc (small letters)
RE: The genes have been typed with lowercase in the revised manuscript.
Line 77 – please define ESBL here
RE: ESBL has been defined in the revised manuscript.
Line 84 – please italicize gene names. Also check this throughout the manuscript (lines 115, 117, 136, 191 etc)
RE: The genes names have been typed italicized in the revised manuscript.
Line 109 – “K. pneumonias” – please correct a typo
RE: The correction has been made to the typing of this word in the revised manuscript.
Table 1 – I suggest moving the table to supplementary materials, if the primer sequences were not obtained by authors and were derived from the literature
RE: Thanks for the suggestion, we have move the table 1 to supplementary material.
Line 161 – figure 2 does not contain phylogenetic tree – it appears in the figure 1. In addition, the figures are not numbered in the order they appear in the manuscript. Probably, the numbers should be swapped. Please fix. The same is true for line 171 (should be figure 2, not 1).
RE: Thanks for pointing the mistakes. We have fixed the number of the figures and the indications in the revised manuscript.
Table 2 – MLST allele information is not necessary – it is common knowledge. The reference to bigSDB database will be enough
RE: Thanks for the suggestion. We would keep allele information in the table as some of the other paper also put this in the table, and could be taken as reference for the readers.
Table 2 – the STs are not sorted numerically. E.g., ST25 should come before ST107 – this will make the table more human-readable.
RE: The order of the STs have been modified in the revised manuscript.
Lines 171-180 and throughout the manuscript – I think that ‘isolates’ would be more appropriate than ‘strains’ in this case because strain definition usually is more specific than the point of isolation. These terms are not synonymous
RE: This has been replaced throughout the paper as suggested.
Line 236 – why the widely prevalent ST was not detected in your study? This fact should be discussed further
RE: The discussion on this point has been added to the revised manuscript.
Line 263 – what was the reason for selection of these particular virulence genes? E.g., genes from mrk cluster or heave metal-resistance genes are also important
RE: We agree with that those genes involved in mrk cluster and heave metal resistant gene are also important, however, for the detection of the virulence regarding the infections of KP to the host, we would be interested in the genes related to capsula, adhesin and siderophores, and selected some of the most prevalent. We believe that mrk may also reflect the isolated KP with indications of adhesin activity. We believe that the studies are needed in the future to investigate as more the genes as possible.
Line 271 – “were completely expressed” – you have not confirmed the gene expression – only its presence. Please fix.
RE: This has been modified in the revised manuscript.

Reviewer 3 Report
Dear authors, the topic chosen for the study presented in this manuscript is very topical and also of great interest to the scientific community, especially since the bacterium chosen is not one that has been studied much from this point of view.
I believe that the experimental design is clear, well organized and the conclusions and discussions are well presented. However, there are some statements for which I have not found bibliographic support, such as from the abstract of the article. In the first sentence, you claim that K. pneumoniae has become the dominant pathogenic bacterium in the case of bovine mastitis. Maybe you should think about reformulating this phrase. I am sure that it is not applicable world wide. Regarding the Material and Method section, I think you were very succinct in your expression, I am not very bothered by this, but, in case of publication, you will have to give more details about the equipment used, for example. I think it would be desirable to improve this section as well and also, to edit the Tables in the manuscript.
Author Response
Dear Editors and Reviewers:
Thank you for your letter and for the reviewers’ comments concerning our manuscript entitled “The characteristics of multilocus sequence typing, virulence genes and drug resistance of Klebsiella pneumoniae isolated from cattle in northern Jiangsu, China”. Those comments are all valuable and very helpful for revising and improving our paper, as well as the important guiding significance to our researches. We have studied comments carefully and have made correction which we hope meet with approval. Revised portion are highlighted in yellow in the paper. The main corrections in the paper and the responds to the reviewer’s comments are as flowing:
Reviewer 3
Dear authors, the topic chosen for the study presented in this manuscript is very topical and also of great interest to the scientific community, especially since the bacterium chosen is not one that has been studied much from this point of view.
I believe that the experimental design is clear, well organized and the conclusions and discussions are well presented. However, there are some statements for which I have not found bibliographic support, such as from the abstract of the article. In the first sentence, you claim that K. pneumoniae has become the dominant pathogenic bacterium in the case of bovine mastitis. Maybe you should think about reformulating this phrase. I am sure that it is not applicable world wide. Regarding the Material and Method section, I think you were very succinct in your expression, I am not very bothered by this, but, in case of publication, you will have to give more details about the equipment used, for example. I think it would be desirable to improve this section as well and also, to edit the Tables in the manuscript.
RE: Thank you very much for giving the valuable comment which makes the paper a great improvement. The sentence mentioned in the Abstract section has been reorganized in the revised manuscript. Also, more detailed information regarding M&M has been added to the revised manuscript.

Round 2
Reviewer 2 Report
The authors have addressed most comments appropriately. However, some minor errors still exist.
Line 106 – should be “sequences”, not “sequencies”
Line 125 – the tree is shown in figure 1, not figure 2
Line 242 - “was mainly identified in humans and is carbapenem resistant, which the dairy farms are not well studied” – please rephrase. It is not clear what is not studied – ST11, carbapenem resistance or whatever. Carbapenem resistance was already mentioned in the previous sentence.
Line 278 – “completely present” – I suggest rephrasing to “present in all isolates” if this is what was meant.
Line 323 – please italicize K. pneumoniae, and give references for the statement of hybrid plasmid existence, e.g., 10.3390/antibiotics9120862; https://doi.org/10.3390/microorganisms7090326, https://doi.org/10.3390/antibiotics10060691, or any other suitable links
Author Response
The authors have addressed most comments appropriately. However, some minor errors still exist.
Line 106 – should be “sequences”, not “sequencies”
RE: This has been corrected in the revised manuscript.
Line 125 – the tree is shown in figure 1, not figure 2
RE: This has been corrected in the revised manuscript.
Line 242 - “was mainly identified in humans and is carbapenem resistant, which the dairy farms are not well studied” – please rephrase. It is not clear what is not studied – ST11, carbapenem resistance or whatever. Carbapenem resistance was already mentioned in the previous sentence.
RE: Thanks, the sentence has been rephrased in the revised manuscript.
Line 278 – “completely present” – I suggest rephrasing to “present in all isolates” if this is what was meant.
RE: Thanks, this has been rephrased in the revised manuscript.
Line 323 – please italicize K. pneumoniae, and give references for the statement of hybrid plasmid existence, e.g., 10.3390/antibiotics9120862; https://doi.org/10.3390/microorganisms7090326, https://doi.org/10.3390/antibiotics10060691, or any other suitable links
RE: Thanks, we have made corrections and added references in this part as suggested in the revised manuscript.